# Preliminary Analysis of Transcriptome Response of *Dioryctria sylvestrella* (Lepidoptera: Pyralidae) Larvae Infected with *Beauveria bassiana* under Short-Term Starvation

**DOI:** 10.3390/insects14050409

**Published:** 2023-04-25

**Authors:** Hongru Guo, Niya Jia, Huanwen Chen, Dan Xie, Defu Chi

**Affiliations:** Key Laboratory for Sustainable Forest Ecosystem Management-Ministry of Education, College of Forestry, Northeast Forestry University, Harbin 150040, China; ghr32170@nefu.edu.cn (H.G.); jianiya9937@126.com (N.J.); chenhuanwen2020@126.com (H.C.); xiedan0807@126.com (D.X.)

**Keywords:** *Dioryctria*, *Beauveria bassiana*, immune system, detoxifying enzyme, antioxidant enzyme

## Abstract

**Simple Summary:**

*Dioryctria* is a destructive borer pest genus that is widely distributed in the trunks, tops, and cones of trees in coniferous forests. It is very difficult to control because it exhibits remarkable stealthiness and its populations develop in an irregular manner. Most *Dioryctria* populations overwinter in the litter layer. As an environmentally friendly pest management practice, the application of *Beauveria bassiana* in the litter may enable the control of pre-wintering pests over a long time period. In this study, we used a wild-type *B. bassiana* strain. Using RNA-Seq data, we detected genes with significantly changed expression in the immune system of *Dioryctria sylvestrella* larvae. In addition, detoxifying enzymes and protective enzymes were also detected. These findings offer molecular-level insights into the interaction of insect and pathogen in the pre-winter period and provide a clear target for improving the effectiveness of *B. bassiana* in controlling attacks by *Dioryctria*.

**Abstract:**

The *Dioryctria* genus contains several destructive borer pests that are found in coniferous forests in the Northern Hemisphere. *Beauveria bassiana* spore powder was tested as a new method of pest control. In this study, *Dioryctria sylvestrella* (Lepidoptera: Pyralidae) was used as the object. A transcriptome analysis was performed on a freshly caught group, a fasting treatment control group, and a treatment group inoculated with a wild *B. bassiana* strain, SBM-03. Under the conditions of 72-h fasting and a low temperature of 16 ± 1 °C, (i) in the control group, 13,135 of 16,969 genes were downregulated. However, in the treatment group, 14,558 of 16,665 genes were upregulated. (ii) In the control group, the expression of most genes in the upstream and midstream of the Toll and IMD pathways was downregulated, but 13 of the 21 antimicrobial peptides were still upregulated. In the treatment group, the gene expression of almost all antimicrobial peptides was increased. Several AMPs, including cecropin, gloverin, and gallerimycin, may have a specific inhibitory effect on *B. bassiana*. (iii) In the treatment group, one gene in the glutathione S-transferase system and four genes in the cytochrome P450 enzyme family were upregulated, with a sharp rise in those that were upregulated significantly. In addition, most genes of the peroxidase and catalase families, but none of the superoxide dismutase family were upregulated significantly. Through innovative fasting and lower temperature control, we have a certain understanding of the specific defense mechanism by which *D. sylvestrella* larvae may resist *B. bassiana* in the pre-wintering period. This study paves the way for improving the toxicity of *B. bassiana* to *Dioryctria* spp.

## 1. Introduction

*Dioryctria* (Pyralididae, Lepidoptera) species are harmful to the cones, trunks, and branches of many Pinaceae plants. In recent years, there have been increasing reports on the damage caused by *Dioryctria* species in coniferous forests of the Northern Hemisphere, in North America [1,2,3], Europe [4], the Russian Far East [5], China [6], and Korea [7]. Among these species, *D. sylvestrella* is especially important. Serious infestations of *D. sylvestrella* have occurred across the northeastern areas of China. Similar to the damage of maritime pine (*Pinus pinaster* Ait.) in France [8] and brutian pine (*Pinus brutia* Ten.) in Turkey [9], its larvae damage Korean pine and Mongolian Scotch pine and cause the death of saplings and the branches of large trees. In Korean pines, just one larva can cause the withering and early falling of single cones (by nibbling) or clusters of cones (by boring in the tips of branches), sharply reducing the yield and quality of seeds, a famous and expensive nut.

In northeastern China, *D. sylvestrella* and *D. abietella* are the main pest species affecting Korean pine [6]. *Dioryctria* larvae overwinter as third- or fourth-instar larvae. In terms of their life history, most of their populations overwinter in wormholes under the bark near the ground or in shallow layers of soil to escape the cold. This period may last for more than 6 months, between October and May. In the early years in India, some scholars tried to use thimet, carbofuran, dimethoate, or monocrotophos to drip-irrigate the soil to control *D. abietella*, which is harmful to *Picea smithiana*, and achieved some effect [10,11]. However, such highly toxic pesticides seriously pollute the environment and threaten biosafety and have been banned by various countries.

The soil is the richest reservoir of microbial species and also contains *Beauveria bassiana*, which parasitizes and kills overwintering or ground-moving insects on the surface, as commonly reported in studies of many orchard pests [12,13]. In support of this idea, in 2020, we found corpses of *Dioryctria* larvae killed by infection with a wild *B. bassiana* strain while overwintering in forest litter. Due to this, spraying the floors of pine-nut orchards with *B. bassiana* spore powder preparation may reduce the survival rate of overwintering populations. Such a practice would also ensure that pesticides were not sprayed directly onto nuts and thus guarantee food safety. This led to our research into the use of *B. bassiana* preparations to control *Dioryctria*.

*B. bassiana* is an entomopathogenic microorganism. According to still-incomplete statistics, it can infect more than 1000 species of hosts [14]. Among the biopesticides most widely used today, almost 40% of the effective ingredients are derived from *B. bassiana* [15,16]. However, it can be resisted by the defense systems of insects. Humoral immunity mainly involves the recognition of pathogen types through peptidoglycan recognition proteins (PGRPs) [17]. Then, by signaling through the Toll and immune deficiency (IMD) pathways, a series of related enzymes are triggered that activate intracellular signaling through extracellular cascades, eventually inducing upregulation in the expression of various antimicrobial peptides (AMPs) by genes in the genome, so that the corresponding secretion of AMPs significantly increases and invading pathogens are inhibited or killed [18]. There are many types of AMPs; most exhibit a broad spectrum of antibacterial activity; however, some exhibit high activity against one or two specific bacteria only [19,20]. In line with previous research, in this study we sought to find highly expressed AMPs, especially antifungal peptides (gallerimycin family, heliomicin family, etc.) [21,22].

*B. bassiana* also affects physiological processes in insects by secreting a variety of secondary metabolites, including beauvericin, bassianin, bassianolide, beauverolides, tenellin, oosporein, and oxalic acid [23]. These toxins facilitate fungal invasion or act as immunosuppressive compounds [24], accelerating insect death and tissue decomposition, and thereby promoting reproduction of the pathogens themselves [25]. In insects, a variety of enzyme families contribute to the metabolic degradation of harmful foreign substances; these include cytochrome P450 enzymes (P450s) [26,27], glutathione-S-transferases (GSTs) [28], the carboxylate enzyme system (CarbEs) [29], the UDP-glucosyltransferase system [30], and ATP-binding cassette transporters (ABCs) [31]. The overexpression (or mutation) of various detoxification-enzyme genes can cause changes in the sensitivity of insects to various types of insecticides. Previous studies have indicated that ABCs play important roles in the insect metabolism of Cry1Ac, a kind of BT toxin [32,33,34]. However, because the secretion of toxins or other secondary metabolites by pathogenic microorganisms is highly complex, there have been few studies on insect detoxification enzymes or on the interactions between insects and pathogenic microorganisms. In the process of inhibiting the reproduction of microbial pathogens, reactive oxygen species (ROS) are secreted in large quantities [35]. Antioxidant enzymes can quickly and effectively scavenge excess ROS while maintaining a steady state and protecting insect tissues from damage by free radicals. The main insect antioxidant enzymes include peroxidase (POD), superoxide dismutase (SOD), and catalase (CAT) [34].

In addition, research into the immune systems and detoxification-enzyme systems of insects has mostly been based on model species such as *Bombyx mori* [36], *Drosophila* [37], or mosquito [38]. There have been few studies on the pest species that affect many cash crops in agriculture and forestry [39,40,41]. In this study, the larvae of *D. sylvestrella* were used as the research object, and the wild *B. bassiana* strain SBM-03 was isolated from cadavers of *D. abietella*, another species of *Dioryctria* found at the same site as our target pest. The next-generation sequencing method was used to conduct a comparative analysis at the transcriptome level, focusing on the expression of immune metabolism, detoxification enzymes, and protective enzymes in larvae, in order to further understand the defense strategies of *D. sylvestrella* against fungal infection.

## 2. Materials and Methods

### 2.1. Acquisition and Culture of Beauveria Bassiana Strain

The *B. bassiana* strain SBM-03 isolated in our laboratory was used in this study. This is a new wild strain that was isolated and purified by the authors from a *D. abietella* corpse found at the Korean pine plantation in Boli County, Qitaihe, Heilongjaing Province, China (45.96° N, 130.30° E) in 2019. This strain was registered at the China General Microbiological Culture Collection Center (CGMCC) with the record number CGMCC No. 18818 in the same year. The strain used in the experiment was a second-generation strain derived from the original strain after isolation. The fungus was routinely propagated using quarter-strength Sabouraud dextrose agar (1/4SDA) and incubated at 28 ± 1 °C in a dark environment. Conidia were harvested from cultures grown for 10 days at 28 °C and suspended in 0.02% Tween-80 (Coolaber Technology Co., Ltd., Beijing, China) [42].

### 2.2. Acquisition and Disposal Method of Dioryctria sylvestrella Larvae

In our observations, it was found that most of the larvae of *D. Sylvestrella* over-wintered in the form of third- or fourth-instar larvae; however, it was rare to see individuals in the second instar at this time. Third-instar larvae are small and do not survive well in breeding. In order to maintain the uniformity of experimental insect individuals, we only took the fourth-instar larvae of *D. sylvestrella* used for the test back to the Northeast Forestry University laboratory.

The fourth-instar larvae of *D. sylvestrella* used for the test were collected from damaged branches of Korean pine in the Lushui River Seed Garden in Changbai Mountain, Jilin Province, China (42.52° N, 127.79° E) and taken back to the Northeast Forestry University laboratory. The treated larvae were placed in a 90 mm petri dish filled with high-temperature-sterilized dry pine needles. Larvae were reared in an incubator at 16 ± 1 °C in a dark environment with a relative humidity of 60 ± 5% (RH). Compared with many other methods of rearing insects at temperatures above 25 °C [43,44,45], our rearing environment is significantly lower. This is a similar environment to the autumn litter of coniferous forests in the Northern Hemisphere. The pests feed on the inner tissue of cones and shoots and do not seem to feed on dried pine needles or soil. The larvae were collected before winter, when they were in a state of preparation for overwintering and had almost stopped feeding.

The total RNA was extracted immediately from 10 freshly captured larvae and set as the control group, CK1. In addition, total RNA was also extracted from another 10 larvae that were starved for 72 h; this was set as short-term starvation experimental group T1.

Furthermore, another 10 larvae were sprayed with 2 mL of 0.02% Tween-80; their total RNA formed the control group CK2. Finally, 10 larvae were sprayed with 2 mL of the conidial suspension (1 × 10^7^ conidia/mL emulsified with 0.02% Tween-80). These inoculated larvae were then reared at 18 ± 1 °C in a dark environment with a relative humidity (RH) of 60 ± 5%. They were also not fed any food for 72 h. Their total RNA extraction was set as experimental group T2. Fresh cadavers with mycosis were separated on a daily basis, examined to confirm the Bb SBM-03 infection, and their details recorded.

The above procedure was carried out three times each for the CK1, CK2, T1, and T2 groups.

### 2.3. RNA Isolation for Sequencing

Whole bodies of larvae were collected individually, ground with liquid nitrogen, and then stored at −80 °C.

Trizol reagent (Invitrogen Life Technologies, Carlsbad, CA, USA) was used according to the manufacturer’s instructions. RNA purity and integrity were quantified using an ASP-2680 spectrophotometer (ACT Gene, Piscataway, NJ, USA) and an Agilent Technologies 2500 Bioanalyzer (Agilent Technologies, Palo Alto, CA, USA). Samples were then used for sequencing library construction.

### 2.4. Next-Generation Sequencing and Raw Data Analysis

Complementary DNA (cDNA) libraries were constructed using an Illumina Truseq™ Sample Prep Kit (Illumina, San Diego, CA, USA). All the samples were sequenced on an Illumina HiSeq^TM^2500 platform (Illumina, San Diego, CA, USA). The original data were stored in FASTQ format [44,46].

### 2.5. Assembly and Annotation of Transcripts

Data in respect of low-quality raw sequencing reads were filtered using the analysis platform of Omicshare (Gene Denovo Biotechnology Co., Ltd., Guangzhou, Guangdong province, China). The specific content included (i) the removal of reads containing adapters; (ii) the removal of reads with a ratio of unknown nucleotides (N) greater than 10%; and (iii) the removal of low-quality reads with a quality value Q ≤ 20 accounting for more than 40% of the entire read. By such means, high-quality, clean reads were obtained.

The high-quality reads were then imported into the Trinity assembler (http://trinitymaseq.sourceforge.net (accessed on 14 August 2019)) to assemble all the clean reads de novo and produce contigs and singletons [47]. By comparison with known sequences in the public database of the Non-RedundantProtein Sequence Database (Nr), the functional annotation of unigenes was achieved. The assembled unigene sequences were annotated against the NCBI non-redundant protein sequence databases using BLASTX (E-value < 10^−5^) [48,49].

Differentially expressed genes (DEGs) were calculated using the reads per kilobase per million reads mapped (RPKM) method, and the screening conditions (|log2fold change| ≥ 1 and *p*-value < 0.05) were used to identify the DEGs [50]. DEGs were classified and annotated using Gene Ontology (GO) term enrichment analysis (http://www.blast2go.com/b2ghome/ (accessed on 14 August 2019)) and Kyoto Encyclopedia of Genes and Genomes (KEGG) pathway enrichment analysis (http://www.genome.jp/kegg/genes.html (accessed on 14 August 2019)) at the threshold of corrected *p* < 0.05. The functional category for DEGs was assigned using the Eukaryotic Orthologous Group (KOG) database (http://www.ncbi.nlm.nih.gov/COG/grace/shokog.cgi (accessed on 14 August 2019)). To predict genes involved in pathogen–host interactions, BLAST searches were conducted against protein sequences in the pathogen–host interaction (PHI) database (version 4.13, http://www.Phi-base.org/ (accessed on 14 August 2019)) (e < 1 × 10^−5^) [51].

### 2.6. Tissue Differential Expression Assessed via qRT-PCR

Another 120 healthy *D. sylvestrella* larvae were selected for a validation experiment. The larvae were treated identically to those described in Section 2.2 above. Ten larvae from each treatment group were randomly chosen, and total RNA was extracted from their whole bodies. Each procedure was carried out in triplicate. Amounts of 1 μg of RNA were reverse-transcribed using the StarScript II First-strand cDNA Synthesis Kit (GenStar, Beijing, China). A quantitative real-time PCR (qRT-PCR) was performed on an CFX96 Touch Real-Time PCR System (Bio-Rad Laboratories Co., Ltd., Hercules, CA, USA) using GoTaq qPCR Master Mix (Promega, Madison, WI, USA) at a 20 µL volume. Each well plate was loaded with 1 μL cDNA. The reaction steps were as follows: 95 °C for 2 min; 40 cycles of 95 °C for 15 s; and 60 °C for 30 s. The data were analyzed using the 2^−ΔΔCt^ method [39,48,52]. The primers were described previously (Appendix A).

### 2.7. Data Analysis

Microsoft Excel 2016 and Graphpad 8.3 were used for statistical analyses and to construct figures. Further, *t*-testing was used to analyze the survival curve, and qPT-PCR data were analyzed using the 2^−ΔΔCt^ method.

## 3. Results

### 3.1. Dioryctria sylvestrella Larvae Mortality after Infection with Bb SBM-03

In our pre-experiment, involving 30 healthy larvae inoculated with Bb SBM-03, the first deaths were recorded on day 4, and 100% mortality was reached on day 12 (Figure 1a). Four days after the fungal inoculum was sprayed, pathogenicity was observed under laboratory conditions. The mycotized larvae had white hyphal masses, and the typical color (milk-white) of sporulation was observed on the larvae’s surface (Figure 1b). In the non-treated control group, mortality was <20%. Hunger and dry conditions can have an impact on death. Our experimental conditions are not as humid as in the actual natural environment, resulting in some larvae dying due to drying in the later stages of the experiment. The mortality rate in the actual natural environment may be smaller than that in the experiment. In addition, although the temperature of the experiment is a lower constant temperature, it is slightly different than the temperature in the natural environment. Some larvae may not be able to enter a lower metabolic level as soon as possible and die of starvation.

### 3.2. Transcriptome Analysis after Starvation and BbSBM-03 Infection

In the present study, the genes upregulated in *D. sylvestrella* fourth-instar larvae infected by Bb SBM-03 were identified. A total of 35,639,209 bp raw reads were generated from the non-infected and infected larvae samples; these were evaluated for quality, and adaptors were removed by data trimming. The total contigs were further clustered into 41,132 DEGs based on shared sequence content.

All sequences were annotated using GenBank Nr, Swissprot, KEGG, and KOG database searches. To identify highly expressed genes in larvae infected with *B. bassiana*, the transcriptomes of T1 and T2 were compared via DEG analysis. Among the newly captured active larvae, the expression levels of most of the unigenes were significantly changed.

Under the screening conditions of |log2 fold-change| > 1 and FDR ≤ 0.05, in the non-infected test group (CK1 vs. T1), 16,969 genes exhibited significant differences in expression. A total of 3834 genes were upregulated and 13,135 genes were downregulated (Figure 2a). In infected larvae subjected to starvation treatment (CK-2 vs. T2), 16,665 genes exhibited significant differences in expression. Of these, 14,558 genes were upregulated and 2107 genes were downregulated (Figure 2b).

The DEGs of the text groups were enriched in GO terms and classified by gene function: biological processes (cellular activity and metabolic processes); molecular functions (catalytic activity and binding); and cellular components (cell part, membrane part, organelle, and protein-containing complex) (Figure 3).

We identified 238 immunity-related genes from 41,629 unigenes, and these were compared with known protein sequences in *Anopheles gambiae* [53], *Drosophila melanogaster* [54], *Bombyx mori* [55], *Lymantria dispar* [56], and *Ectropis obliqua* [39]. The immunity-related genes were classified on the basis of functions into categories of recognition, modulation, transduction, and effectors. Transduction genes included members of the toll, IMD, JNK, and JAK/STAT pathways (Figure 4, Appendix A). All transcriptome data were classified using the KEGG database and related genes belonging to the toll (32 unigenes), IMD (52 unigenes), JNK (7 unigenes), and JAK/SATA (4 unigenes) pathways. The changes and specific numbers of these genes in test groups T1 and T2 are shown in Figure 5.

On the other hand that we are concerned about, the changes of detoxification- and protective-enzymes genes in the fourth instar larvae of *D. sylvestrella* in test groups T1 and T2 are shown in Table 1.

### 3.3. Expression Patterns of Key Related Genes Verified by qRT-PCR

Ten key genes with different expression patterns in the transcriptome were verified by qRT-PCR, including seven immune-related genes, one cytochrome P450 enzyme gene, one glutathione S-transferase gene, and one peroxidase gene (Figure 6). After starvation for 72 h only, the expression levels of six of these ten genes—PGRP-3, SPH-18, SPH-34, Jun, P450-52, and GST-2—were significantly downregulated in the experimental group compared with the control group. The immune response effector genes Gloverin-2 and Moricin-4 were still significantly increased in the state of starvation only, while the expressions of Cecropin-1 and POD-14 genes in the normal state and after starvation for 72 h remained at very low levels and did not change significantly. After inoculation with *B. bassiana* and starvation for 72 h, except for the immune-related gene Jun, which was slightly downregulated, the expression of the other nine genes was significantly increased. This is basically consistent with the results obtained with respect to the transcriptome (Appendix A).

## 4. Discussion

In this study, RNA-seq was used to analyze changes in the transcriptome expression of whole larvae subjected to short-term fasting or to fasting following infection with *B. bassiana*. In many studies, hemocytes and fat bodies are tested separately [39,56]. However, in our assessment, a transcriptome of the whole larva body can better reveal any comprehensive changes in the overall metabolic and physiological processes of larvae.

In the case of fasting for 72 h, the expression of 16,969 genes was significantly down-regulated while the expression of 3834 genes was increased; under the condition of *B. bassiana* infection, the expression of 14,558 genes was significantly increased while the expression of 2107 genes decreased. Apparently, the infection with pathogenic microorganisms changed the overall metabolic process—which should have entered a dormant state—from downregulation to upregulation. Under fasting conditions and infected by *B. bassiana*, an elevated metabolic process accelerated the depletion of stored nutrients. We predict that this could lead to increased mortality in overwintering pests, even if they do not die from *B. bassiana* infection.

In terms of transcriptome changes affecting the humoral immunity of the larvae, fasting for 72 h increased the expression of 83 functional genes related to the immune response and decreased the expression of 145 genes. The genes with increased expression included multiple effector molecule genes (AMPs), indicating that short-term fasting improved the ability of larvae to resist foreign microorganisms. In insects, many basic studies have proved that the synthesis of energy-storage substances such as glycogen or fat, as well as many immune function molecules, is completed in fat bodies [57], and that the two physiological processes have multiple identical regulatory factors or key molecules [58]. For example, the two physiological processes both stimulate the body to secrete the same hormone, octopamine (OA), which can act on the G-protein-coupled receptors of the fat body and promote the oxidation of energy storage substances in cells, thereby supplying energy [59,60]. Both processes lead to a decline in insulin-like signaling pathways (ILSs), thereby inhibiting the synthesis and accumulation of triglycerides [61,62]. Similarly, the activation of toll receptors caused by pathogen infection and starvation can both inhibit the phosphorylation of AKT and lead to the activation of the transcription factor FOXO [63]. The physiological metabolism associated with the activation of FOXO is very complex but ultimately leads to a reduced growth rate and developmental delay in insects. These both lead to more energy being invested, either in response to starvation or in resistance to the pathogen, and this affects survival rates [57]. The results of this experiment support the above point of view. However, we need to exclude some effector molecules whose expression was upregulated only by starvation alone because it is not easy to judge whether these have any specific effects in cases of *B. bassiana* infection.

Under the conditions of 72 h fasting and *B. bassiana* infection, the number of upregulated immune-related genes increased to 161, while the number of downregulated genes decreased to 56. The expression levels of most genes related to humoral immunity in larvae, including immune recognition proteins, signal transduction molecules, and various antimicrobial peptides (AMPs), were increased. Previous research into model organisms such as *Bombyx mori* or *Galleria mellonella* has found that specific immune recognition proteins, information transmission of toll and IMD signaling pathways, and the secretion of several AMPs [64,65,66,67,68,69] all play major roles in the processes by which Lepidoptera resist *B. bassiana* infection. In this experiment, after excluding the genes whose expression was significantly increased under short-term fasting, three PGRPs were screened out (Unigene0009929, Unigene0015572, and Unigene0025313). Among AMPs, one gloverin (Unigene0014980), four cecropins (Unigene0003409, Unigene0005574, Unigene0024777, and Unigene0024779), and one gallerimycin (Unigene0015087) may be specifically involved in inhibiting the reproduction of *B. bassiana* and killing larvae. Furthermore, in many insects, the Duox-ROS pathway is highly active during infection. The reproduction of microbial pathogens in insects stimulates the activation of the Duox-ROS system; this promotes the secretion of active oxygen components, which can directly change the homeostasis of insects; indeed, it can inhibit the reproduction of almost all microorganisms [70,71]. For this reason, inhibiting the expression of Duox-ROS is expected to become an effective means of improving the virulence of microbial pesticides [72].

The expression of genes related to the detoxification enzyme systems of larvae was different after fasting for 72 h. Compared with larvae treated with fasting only, insects infected with *B. bassiana* produced significantly increased numbers of four cytochrome P450 enzymes (Unigene0028402, Unigene0030240, Unigene 0019662, and Unigene 0015555) and one glutathione-S transferase enzyme (Unigene0034104). These results were similar to those obtained in previous pathogen infection tests involving *Aphis gossypii* [68] and *Plutella xylostella* [69]. It is predicted that these detoxification enzymes may be involved in the metabolism of larvae against toxins secreted by *B. bassiana.*

In the case of fasting for 72 h, only the expression of two or three peroxidases (PPOs) increased significantly, and the expression of other antioxidant protection enzymes did not change greatly. The expression of catalases (CATs) and of almost all kinds of PPOs increased significantly after infection with *B. bassiana*. The high expression of such antioxidant enzymes is directly related to the activation of the Duox-ROS system. Antioxidant enzymes, especially CATs, remove excess ROS and prevent excessive damage to body cells [51]. The results relating to PPOs and CATs were in line with expectations. However, in our experiments, the expression of the whole superoxide dismutase (SOD) family exhibited little change.

In this study, we used a single time-point of 72 h to analyze metabolic changes in larvae transcriptomes. This is because, after 96 h, the larvae of the experimental group infected with *B. bassiana* began to die. We simulated the living conditions of insects in autumn, i.e., prior to overwintering; however, the ambient mean temperature (about 18 °C) of our testing location was lower than that in many similar studies (typically above 24 °C). Such a difference in temperature affects the metabolism of larvae and also the reproduction rate of *B. bassiana*. In addition, the sampling time in the current study was prolonged compared with similar experiments involving other insects. Nonetheless, our experiment produced satisfactory results overall. For insects of the genus *Dioryctria*, a group of immune system genes were screened. We identified detoxification-enzyme and antioxidant-enzyme genes involved in the resistance of larvae to *B. bassiana* infection before overwintering, and effects due to short-term starvation were excluded. These results improve our understanding of the physiological response of insects to starvation stress and of interactions between hosts and pathogenic microorganisms.

## 5. Conclusions

Through transcriptome sequencing and a comparative analysis of bioinformatics, using the real-world situation prior to overwintering as a reference, we investigated changes in transcriptome expression in fourth-instar larvae of *D. sylvestrella* under conditions of short-term starvation and *B. bassiana* infection. To some extent, this ruled out the impact of short-term fasting in the early winter. We observed changes in the physiological processes of larvae under both conditions, and we paid special attention to the humoral immune system and to the families of detoxifying and protective enzymes. In addition, we screened and obtained specific and highly expressed genes found in the humoral immune system, as well as several specific and highly expressed genes in the families of P450 cytochrome oxidase and glutathione S-transferase. These findings lay the foundation for a better understanding of changes in the physiological metabolism of insects in the early winter and of their interaction with *B. bassiana*. This may promote the development of new, higher-virulence biopesticides in the future.

## Figures and Tables

**Figure 1 insects-14-00409-f001:**
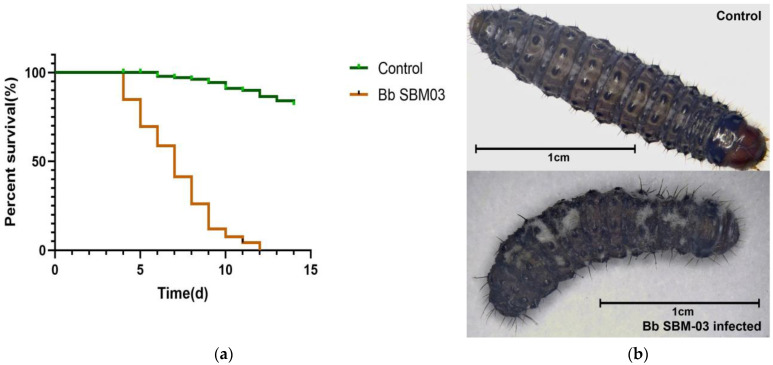
Virulence of BbSBM-03 against *Dioryctria sylvestrella* 4th-instar larvae. (**a**) Percentage of live larvae after spraying with *B. bassiana* conidial suspension (1 × 10^7^ conidia/mL) under laboratory conditions (10 larvae per dish and three replicates). (**b**) A healthy larva and the cadaver of a larva infected with BbSBM-03 (7 days after inoculating).

**Figure 2 insects-14-00409-f002:**
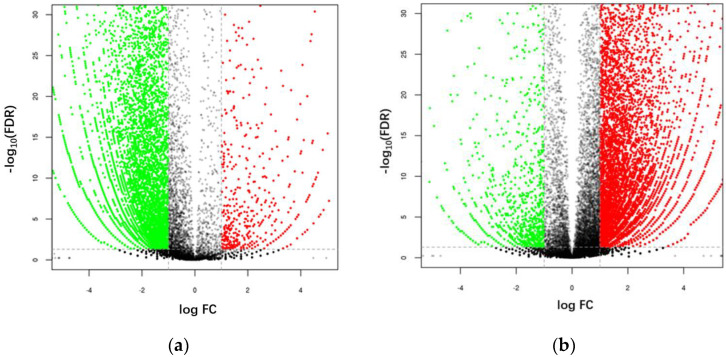
Volcano plot overview of the fold-change (−log_10_ (FDR) > 1) distribution of DEGs. (**a**) Expression levels of the contigs in non-infected larvae group under conditions of short-term starvation only (CK1 vs. T1). (**b**) Expression levels of the contigs in larvae infected with Bb SBM-03 under conditions of short-term starvation (CK2 vs. T2). Green points: Down-regulated genes; Red points: Up-regulated genes; Black points: Genes with insignificant changes in expression. Dot line: threshold to decide whether the genes expression level changes observably.

**Figure 3 insects-14-00409-f003:**
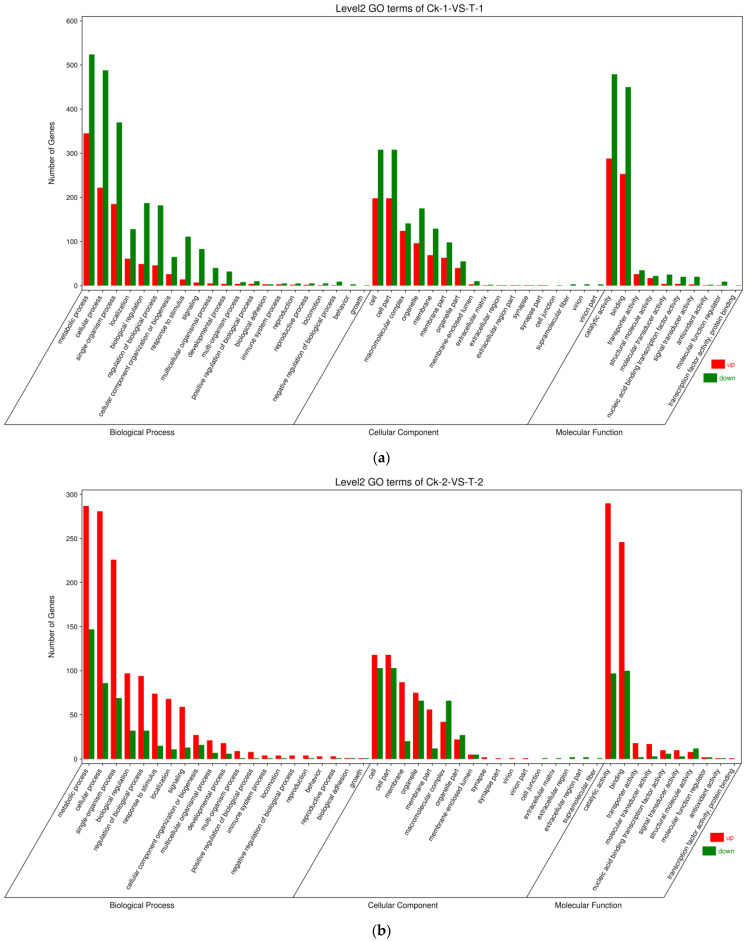
Gene Ontology (GO) annotation of DEGs in the *D. sylvestrella* transcriptome. (**a**) Larvae only in a state of starvation; and (**b**) Larvae in a state of starvation and infection with *Beauveria bassiana*.

**Figure 4 insects-14-00409-f004:**
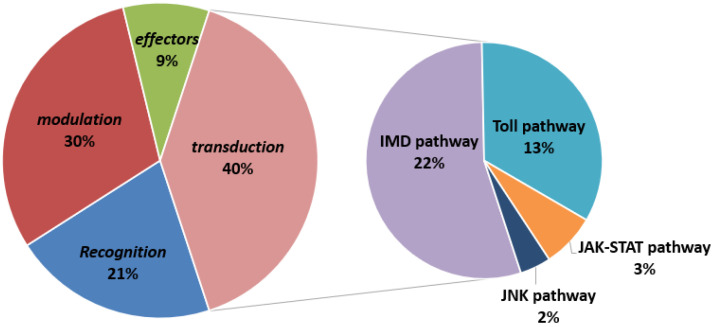
*D. sylvestrella* larvae immunity-related genes in the categories of recognition, signal modulation, transduction, and immune effectors. Transduction include Toll pathway, IMD pathway. Jak-stat pathway and JNK pathway.

**Figure 5 insects-14-00409-f005:**
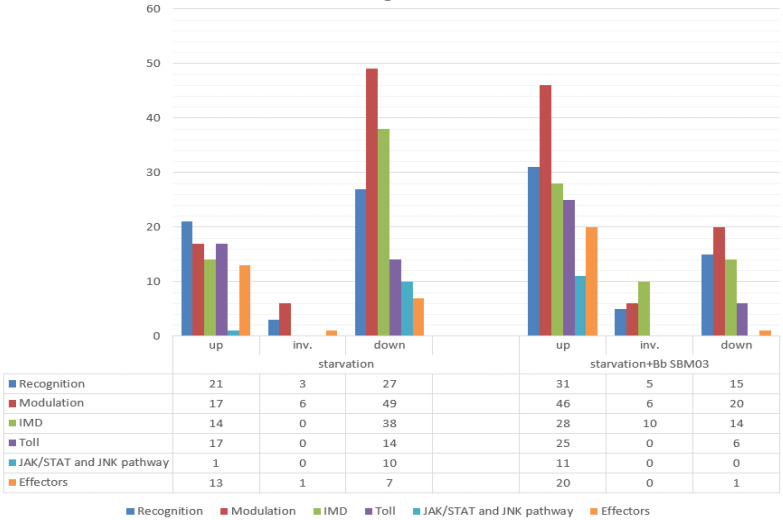
Changes in the expression of immune-related genes of *D. sylvestrella* larvae subjected to starvation only or starvation plus inoculation with Bb SBM-03.

**Figure 6 insects-14-00409-f006:**
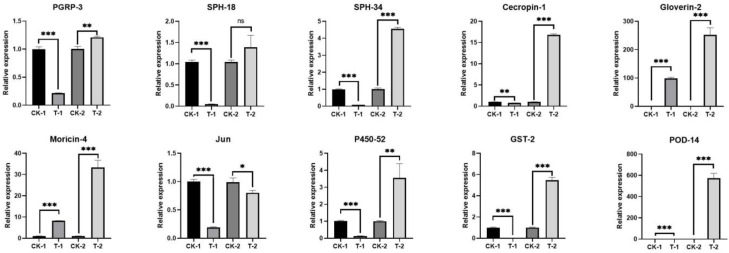
RT-qPCR analysis of expression of ten genes from *D. sylvestrella* larvae subjected either to starvation treatment alone or starvation plus inoculation with Bb SBM-03. The 10 genes include 7 immunity-related genes, 2 detoxifying-enzyme genes, and one protective-enzyme gene. Elongation factor-1 alpha (EF1-α) was used as an internal reference gene. The data are represented as the mean ± S.D. (n = 3). * *p* < 0.05; ** *p* < 0.01; *** *p* < 0.001. ns: no significance.

**Table 1 insects-14-00409-t001:** Changes in the expression of detoxification- and protective-enzyme genes of *D. sylvestrella* larvae subjected to starvation only or starvation plus inoculation with Bb SBM-03.

Gene Groups	Gene Families	Processing Method						Total Number of Genes
		Starvation			Starvation + Bb SBM-03			
		Up.	Inv.	Down.	Up.	Inv.	Down.	
Detoxification enzymes	Cytochrome P450s (P450s)	46	5	59	56	19	35	110
	Glutathione S-transferases (GSTs)	13	2	17	13	5	14	32
	ATP-binding cassette transporters (ABCs)	23	5	39	44	18	5	67
	Carboxylesterases (CarbEs)	12	0	12	8	6	10	24
	UDP-glucosyltransferases (UGTs)	2	0	4	4	1	1	6
Protective enzymes	Superoxide dismutases (SODs)	10	0	5	5	7	3	15
	Catalases (CATs)	4	2	5	7	3	1	11
	Peroxidases (PODs)	6	3	7	15	1	0	16

Up.: |log2 fold-change| > 1. Inv.: −1 < |log2 fold-change| < 1. Down.: |log2 fold-change| < −1.

## Data Availability

The data presented in this study are available upon request from the corresponding author.

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
