# Peer review of "Preliminary Analysis of Transcriptome Response of Dioryctria sylvestrella (Lepidoptera: Pyralidae) Larvae Infected with Beauveria bassiana under Short-Term Starvation"

_insects, 2023, doi:10.3390/insects14050409_

Round 1
Reviewer 1 Report
Dear,
The manuscript "Preliminary analysis of transcriptome response of Dioryctria sylvestrella (Lepidoptera: Pyralidae) larvae infected with Beauveria bassiana under short-term starvation" has scientific relevance, mainly because it addresses the species D. sylvestrella as the target of the study. However, a more important aspect is given in the Introduction to the occurrence of the insect in China, which reduces the impact on publication and also reduces innovation in the use of B. bassiana.
The same considerations can be made about the discussion.
The manuscript is well written, but it would be interesting to be revised to adjust terms most used in scientific writing worldwide.
Author Response
Dear reviewer,
Thank you very much for your patient evaluation and valuable comments on our paper! We have modified part of the text and description in the article according to the other two reviewers` requirements and suggestions.
Reviewer 2 Report
In my opinion, the research is very interesting, showing a great mastery of molecular techniques. The work is very well planned, described, and with results that validate the objectives. In the attached file I have indicated some formal indications and others that could improve the presentation of the work. The article can be published.

English quality is fine. In my opinion, it doesn't need any improvement and can be published in this way.
Author Response
Dear reviewer,
Thank you very much for your patient evaluation and valuable comments on our paper! We have modified part of the text and description in the article according to your requirements and suggestions. The following are specific changes to each recommendation and existing questions.
The abstract has been extensively revised, but it is difficult to summarize the results in about 200 words.
1. Lines 9-38 English words have been changed.
2. Lines 25-36 Yes, these three groups abstractly describe the most important outcomes we care about.
3.Line 59-74 Another reviewer stressed the need to increase references to traditional chemical insecticides to control this pest. The paragraph adds references to previous control methods of the pest using conventional chemical pesticides.
4.Line 77-78. We think it necessary to express the main points of this study in brief paragraph in the introduction?
5.Line 95 Paragraphs have been consolidated.
6. Line 115-135 Revise all punctuation, necessary spaces and prepositions.
7. Lines 161 (ii) remove more than 10% of the unknown nucleotide (N);
8.Line 266-269 There are too many columns in the row bar chart. We may need to enlarge the picture. Is that OK?
9.Line 275 RT-qPCR was used to test the quality and authenticity of transcriptome sequencing. We think there is necessary to add other new validation genes.
10. Comments under the line 259 has been added.
Reviewer 3 Report
Comments to authors
The present study is a significant contribution in the field of biological control of micro-organisms. Exploration of these microbes may be helpful for future food security and safety issues. The authors have implemented best possible ways to meet the objectives of studies. I have some suggestions for the improvement of the manuscript.
Abstract:
Methodology portion of abstract may be increased to attract the readers of this manuscript.
Introduction
1. Line 48-53 need citations.
2. Line 59 you must first discuss about the use of other insecticides for this pest first if any in the previous studies then come to B. bassiana.
3. Line 66-95 relevant examples from previous studies (with typical examples of insect species) may be included to support these two paragraphs
4. Line 100-102 examples from these studies may be included to support line 66-95 in the form of short review
Materials and Methods
As a whole I suggest to compare these studies with any standard insecticides too as such studies are always compared with the use of insecticides too. If possible kindly compare the results by applying any insecticide now a day’s being utilized against this insect pest.
1. Line 117- space issue here China(45.96°N,130.30°E).
2. Line 117 -119 need citation
3. Line 122-143 need relevant citations for
methodology.
4. Line 154 -157 need citation
5. Line 173, 177 have space issues
6. Line 183-191 methodology need citations
7. Line 112 how much older this strain was being maintained please add information.
8. Line 122 why only 4th instars why not second and third instars
Results
1. Line 205. Hunger and dry conditions can have an impact on death. If such situation is always there then you must add something in introduction related with effect of Hunger and dry conditions on insect death in the form of short paragraph.
2. Figure 3 (a,b both) not clear. Make is clear for readers
Discussion
1. Page number 10 need relevant examples of similar studies already done
Conclusion
1. Line 382-387 need to be moved to results somewhere. If possible may be removed
2.
General Remarks
Overall manuscript is written well and with slight modification of methodology would be helpful for the net results of the studies.
Slight improvement in language is suggested
Author Response
Dear reviewer,
Thank you very much for your patient evaluation and valuable comments on our paper! We have modified part of the text and description in the article according to your requirements and suggestions. The following are the specific amendments and existing doubts for each suggestion.
The abstract has been extensively revised,
- Line 48-53. References have been cited.
- Line 59. This is a very difficult to control forest boring pest. I'm sorry to say that research papers on prevention and control methods are almost impossible to find. We cited references to Indian literature on the control of another similar pest of the same genus, Dioryctria abietella.
- Line 66-95. There are already many citations. Where do I need to add?
- Materials and Methods :In northeast China, the area of pine is very large. These forests are not orcharded. Food from forests is harvested only in the fall, rather than being orcharded. many forest contractors in the past few decades, including many now, have not applied pesticides to large areas of forest. In recent years, insect infestation has become more and more serious so a few forest contractors begin spraying insecticides and there are no ideal statistics. In the above we have added references to drip irrigation of soil using highly toxic pesticides in India as suggested by you.
- Line 117 Space have been corrected ï¼›Line 117-119 Citation has been added. Line 122-143 Explain and citation has been added.
- Line 154-157 Citation has been added.
- Line 173-177 space issues have been corrected
- Line 183-191 There are already 3 citations. Where should we need to add?
- Line 112 The strain information have been added.
- Line 122 The reasons for selecting the 4th instar larvae have been explained.
- Results The cause of death in the control group has been explained.
- Line 236-239 There are so many columns in the bar chart that we may have to enlarge the picture. Is that OK?
- Line 255-256 I am sorry that we didn't understand how to modify Figure 5?
- Discussion We think there are already enough citations. Where should we need to add citations?
-
Conclusion Line 382-387 have been removed.